# Associations of Preterm Birth with Dental and Gastrointestinal Diseases: Machine Learning Analysis Using National Health Insurance Data

**DOI:** 10.3390/ijerph20031732

**Published:** 2023-01-18

**Authors:** In-Seok Song, Eun-Saem Choi, Eun Sun Kim, Yujin Hwang, Kwang-Sig Lee, Ki Hoon Ahn

**Affiliations:** 1Department of Oral and Maxillofacial Surgery, Korea University College of Medicine, Korea University Anam Hospital, Seoul 02841, Republic of Korea; 2Department of Obstetrics and Gynecology, Korea University College of Medicine, Korea University Anam Hospital, Seoul 02841, Republic of Korea; 3Department of Gastroenterology, Korea University College of Medicine, Korea University Anam Hospital, Seoul 02841, Republic of Korea; 4Department of Statistics, Korea University College of Political Science & Economics, Korea University Anam Hospital, Seoul 02841, Republic of Korea; 5AI Center, Korea University College of Medicine, Korea University Anam Hospital, Seoul 02841, Republic of Korea

**Keywords:** preterm birth, gastrointestinal disease, machine learning, population data

## Abstract

Background: This study uses machine learning with large-scale population data to assess the associations of preterm birth (PTB) with dental and gastrointestinal diseases. Methods: Population-based retrospective cohort data came from Korea National Health Insurance claims for 124,606 primiparous women aged 25–40 and delivered in 2017. The 186 independent variables included demographic/socioeconomic determinants, disease information, and medication history. Machine learning analysis was used to establish the prediction model of PTB. Random forest variable importance was used for identifying major determinants of PTB and testing its associations with dental and gastrointestinal diseases, medication history, and socioeconomic status. Results: The random forest with oversampling data registered an accuracy of 84.03, and the areas under the receiver-operating-characteristic curves with the range of 84.03–84.04. Based on random forest variable importance with oversampling data, PTB has strong associations with socioeconomic status (0.284), age (0.214), year 2014 gastroesophageal reflux disease (GERD) (0.026), year 2015 GERD (0.026), year 2013 GERD (0.024), progesterone (0.024), year 2012 GERD (0.023), year 2011 GERD (0.021), tricyclic antidepressant (0.020) and year 2016 infertility (0.019). For example, the accuracy of the model will decrease by 28.4%, 2.6%, or 1.9% if the values of socioeconomic status, year 2014 GERD, or year 2016 infertility are randomly permutated (or shuffled). Conclusion: By using machine learning, we established a valid prediction model for PTB. PTB has strong associations with GERD and infertility. Pregnant women need close surveillance for gastrointestinal and obstetric risks at the same time.

## 1. Introduction

Preterm birth (PTB), a delivery occurring between 20^0/7^ and 36^6/7^ gestational weeks, is one of the major unsolved problems of obstetrics. PTB is a main cause of serious neonatal morbidity and mortality, which leads to heavy socioeconomic and public health costs [1,2]. The economic cost of caring for preterm infants in the United States was reported to be 25 billion dollars incrementally [3]. In addition, PTB also affects the long-term health and well-being in the later life of preterm infants [4]. In Korea, the cumulative medical cost for preterm children was approximately 43 million dollars for preterm children during 6 years after discharge from the neonate intensive care unit (NICU) [5]. Even with continual efforts to reduce and prevent PTB, PTB still accounts for approximately 11% of total births globally. Even though more than half of PTB occurs in low- and middle-income countries, the PTB rate is also increasing in higher-income countries and remains as a significant social and medical issue over decades [6].

To fully identify the pathophysiology of PTB is challenging because it is a complex syndrome in which many factors are involved. Studies have reported various risk factors for PTB, including sociodemographic, lifestyle, genetic, medical, and obstetrics factors. Periodontal disease, such as gingivitis, which more than 40% of the population in the United States suffer from, is one of the contributing factors of PTB [7,8,9]. The prevalence of periodontitis in Korea was approximately 24%, and it is reported that periodontitis shares some risk factors with PTB, such as low socioeconomic status, obesity, and smoking [5,10,11]. Gastroesophageal reflux disease (GERD), a common gastrointestinal disease during pregnancy, accompanies with dental disease, including periodontitis and dental erosions [12,13]. The prevalence of GERD worldwide was 13%. Its prevalence has more than doubled in Asia over the past two decades, from 6.0% to 15.0% [14,15]. Based on this linkage between PTB, periodontitis, and GERD, a few studies were conducted to assess the effect of both periodontitis and GERD on PTB. These previous studies found that GERD is one of the major determinants of PTB and has a stronger association with PTB than the association between periodontitis with PTB [16,17]. These studies were clinically meaningful in referring to the importance of GERD in the context of preventing PTB, which is often overlooked. However, previous study studies have limitations regarding either a small number of the study population (731 participants) or a relatively lower range of the area under the receiver-operating-characteristic curves (0.51–0.57) [16,17].

To overcome those limitations, we aimed to establish a high-performance prediction model with machine learning and large-scale population data. Furthermore, we assessed the association of PTB with more various dental diseases than in previous studies.

## 2. Materials and Methods

### 2.1. Participants and Variables

Population-based retrospective cohort data came from Korea National Health Insurance claims for 172,462 primiparous women aged 25–40 and delivered in 2017. This retrospective cohort study was approved by the Institutional Review Board (IRB) of Korea University Anam Hospital on 5 November 2018 (2018AN0365). Informed consent was waived by the IRB.

The dependent variable was PTB (birth before 37 weeks of gestation) in 2017. Four categories of PTB were introduced according to the ICD-10 Code: (1) PTB 1—PTB with premature rupture of membranes (PROM) only; (2) PTB 2—preterm labor and birth without PROM; (3) PTB 3—PTB 1 or PTB 2; (4) PTB 4—PTB 3 or other indicated PTB (Appendix A). The 188 independent variables covered the following information: (1) demographic/socioeconomic determinants in 2016, including age and socioeconomic status measured by an insurance fee with the range of 1 (the highest socioeconomic group) to 20 (the lowest socioeconomic group); (2) dental diseases for any of the years 2002–2016, i.e., dental cavity, oral mucositis, periodontitis, salivary gland disease, tooth loss; (3) gastrointestinal diseases for any of the years 2002–2016, i.e., Crohn’s disease, gastroesophageal reflux disease (GERD), irritable bowel syndrome, ulcerative colitis; (4) obstetric history in 2016, that is, infertility; (5) medication history in 2016 including benzodiazepine, calcium channel blocker, nitrate, progesterone, sleeping pill, and tricyclic antidepressant. The selection of these 186 independent variables were based on previous studies and data availability. These data on disease and medication history were screened from ICD-10 and ATC codes, respectively (Appendix A).

### 2.2. Analysis

Logistic regression and the random forest were used for the prediction of PTB [16,17]. A random forest is a group of decision trees with a majority vote on the dependent variable. The random forest with 100 decision trees and default parameters (GINI criterion, max depth none, max features square root) were employed in this study: 100 training sets were sampled with replacements, 100 decision trees were trained with the 100 training sets, the 100 decision trees made 100 predictions, and the random forest took a majority vote on the dependent variable. The data of 124,606 cases with full information were split into training and validation sets with an 80:20 ratio (99,685 vs. 24,921 cases). Criteria for the validation of the trained models were accuracy (a ratio of correct predictions among 24,921 cases) and the area under the receiver-operating-characteristic curve (the plot of sensitivity vs. 1-specificity). In other words, accuracy can be expressed as the true positive plus the true negative divided by all cases. Likewise, the area under the receiver-operating-characteristic curve can be interpreted as “how much sensitivity can be secured when the threshold of sensitivity rises from 0 to 1 and specificity rises from 0 to 1” [11,12]. Random forest variable importance was introduced for identifying major determinants of PTB and testing its associations with dental and gastrointestinal diseases, medication history, and socioeconomic status. The package “sklearn 1.2.0” in Python (CreateSpace: Scotts Valley, 2009) was employed for the analysis from 15 December 2021–31 July 2022.

## 3. Results

Descriptive statistics for the 124,606 participants are presented in Appendix A. The proportion of those with preterm birth (PTB4) was 5.8% (7285/124,606) in 2017. The mean of socioeconomic status was 11.11 for preterm birth vs. 11.08 for term birth, whereas the mean of age was 32.1 for preterm birth vs. 31.8 for term birth (*p*-value < 0.01). The proportions of those with GERD during 2011–2016 were higher for preterm birth than for term birth, and these differences were statistically significant: 7.7% vs. 6.6% (2011 GERD), 8.2% vs. 7.6% (2012 GERD), 8.9% vs. 8.0% (2013 GERD), 9.8% vs. 9.1% (2014 GERD), 10.5% vs. 9.6% (2015 GERD), and 11.7% vs. 10.0% (2016 GERD). Likewise, the proportions of those with progesterone, tricyclic antidepressant, and infertility in 2016 were higher for preterm birth compared to term birth, and these differences were statistically significant: 19.0% vs. 15.8% (2016 progesterone), 10.8% vs. 9.6% (2016 tricyclic antidepressant), and 27.7% vs. 18.1% (2016 infertility). The findings of the univariate analysis above confirm the positive associations of preterm birth with GERD during 2011–2016, as well as age, progesterone, tricyclic antidepressant, and infertility in 2016.

In Table 1, the random forest with oversampling data registered an accuracy of 84.03, and the areas under the receiver-operating-characteristic curves with the range of 84.03–84.04. Its logistic-regression counterparts were within the ranges of 50.45–60.25 (accuracy) and 54.40–60.19 (areas under the receiver-operating-characteristic curves). The performance measures of the random forest with oversampling data were far beyond those of logistic regression with oversampling data. Here, oversampling is an approach to match the sizes of two groups (participants with and without preterm birth) so that the training of machine learning models can be balanced between the two groups. Based on random forest variable importance with oversampling data in Table 2 and Figure 1, PTB 4 has strong associations with socioeconomic status (0.284), age (0.214), year 2014 GERD (0.026), year 2015 GERD (0.026), year 2013 GERD (0.024), progesterone (0.024), year 2012 GERD (0.023), year 2011 GERD (0.021), tricyclic antidepressant (0.020) and year 2016 infertility (0.019). For example, the accuracy of the model will decrease by 28.4%, 2.6%, or 1.9% if the values of socioeconomic status, year 2014 GERD, or year 2016 infertility are randomly permutated (or shuffled). Among the top 10 important variables, socioeconomic status and age are the well-known contributing factors for PTB [18,19,20]. Infertility was strongly associated with the increased risk of PTB. This finding was consistent with the previous studies which reported the increased risk of PTB in women with a history of fertility or assisted reproductive technology [21,22,23]. GERD during 2011–2016 ranked top 10 important variables. Even though GERD was not considered as the conventional risk factor for PTB, recent studies have reported the association between GERD and PTB [16,17,24]. The mechanism by which GERD and infertility increase the risk of PTB is not elucidated yet. Among the administration of medication, TCA and progesterone acted as important variables. It is assumed that progesterone showed a strong association with PTB, not because it increased the risk of PTB, but because it is widely used to prevent PTB in pregnant women with short cervical length, the high-risk group for PTB. It is reasonable that the administration of TCA, a popular anti-depressant, was associated with the risk of PTB, considering that maternal stress is also a well-known contributor to PTB [25]. In addition, recent studies also demonstrated that antenatal maternal depression increased the risk of PTB [26,27]. It should be noted that the random forest variable importance measures for oversampling data were very similar to those for original data in Appendix A and Figure 1.

## 4. Discussion

### 4.1. Summary

The random forest with oversampling data registered an accuracy of 84.03, and the areas under the receiver-operating-characteristic curves with the range of 84.03–84.04. Based on random forest variable importance with oversampling data, PTB has strong associations with socioeconomic status, age, year 2014 GERD, year 2015 GERD, year 2013 GERD, progesterone, year 2012 GERD, year 2011 GERD, tricyclic antidepressant, and year 2016 infertility.

### 4.2. Contributions

This study presents the most comprehensive analysis of the determinants of PTB, using a population-based cohort of 124,606 participants and the richest collection of 186 predictors, including demographic/socioeconomic determinants, dental and gastrointestinal diseases, and medication history. We established a valid prediction model for PTB and investigated its associations with dental and gastrointestinal diseases. Moreover, this study made the following clinical and policy implications. Firstly, the findings of this study agree with those of previous studies on the positive associations of preterm birth with low socioeconomic status [20,28]. The odds ratio of low socioeconomic status was 1.75 in a retrospective cohort study of 1,282,172 pregnant women in Scotland during 1980–2000 [20]. The corresponding statistic was 5.1 in a multicenter prospective study of 2645 pregnant women in Korea [28]. In a similar vein, low socioeconomic status, measured by an insurance fee with the range of 1 (the highest socioeconomic group) to 20 (the lowest socioeconomic group), ranked first in random forest variable importance and its mean was higher for preterm birth (11.11) than for term birth (11.08) in this study. Secondly, the results of this study request due attention to the importance of infertility and its determinants in the prediction of preterm birth. In this study, the variable importance of the year 2016 infertility was within the top 10, and its proportion was higher for preterm birth (27.7%) than for term birth (18.1%) with statistical significance. These findings are consistent with those of a retrospective cohort study of 2034 pregnant women in Australia during 1986–1998, reporting that the proportion of preterm birth was higher with high infertility treatment history (5.2%) than without the history (1.0%) [22]. Likewise, in a retrospective cohort study of 117,401 pregnant women during 1991–2016 in the United Kingdom, the odds ratios of preterm birth were higher with the following causes of infertility than without the conditions, i.e., ovulatory disorders (1.25), tubal disorders (1.25) and endometriosis (1.17) [23]. However, little literature is available and more analysis is needed on the determinants of infertility and their associations with preterm birth.

Thirdly, the findings of this study affirm those of existing literature on the positive associations of preterm birth with GERD [16,17,24]. In this study, GERD during 2011–2016 ranked within the top 10 in random forest variable importance for the prediction of preterm birth in 2017, and their proportions were higher for preterm birth than for term birth with statistical significance: 7.7% vs. 6.6% (2011 GERD), 8.2% vs. 7.6% (2012 GERD), 8.9% vs. 8.0% (2013 GERD), 9.8% vs. 9.1% (2014 GERD), 10.5% vs. 9.6% (2015 GERD), and 11.7% vs. 10.0% (2016 GERD). Similarly, in a retrospective cohort study of 405,586 pregnant women during 2002–2017 in Korea, the random forest variable importance rankings of GERD during 2009–2014 were within the top 10 for the prediction of preterm birth during 2015–2017 [24]. However, pregnant women usually neglect the significant role of GERD symptoms in preterm birth; hence more active counseling is really needed for effective prenatal care. Fourthly, this study brings new insights into a positive relationship between inflammatory bowel disease and preterm birth. Inflammatory bowel syndromes (IBS) in 2005 and 2006 ranked within the top 20 in random forest variable importance, and their proportions were higher for preterm birth than for term birth in this study, i.e., 3.61% vs. 2.89% (2005 IBS), 3.69% vs. 3.13% (2006 IBS). As a matter of fact, inflammatory bowel disease, including Crohn’s disease and uncreative colitis, were reported to be the risk factors of preterm birth in previous studies [29,30,31,32,33], and inflammatory agents are expected to mediate uterine contractility, cervical dilation, and inflammatory bowel disease during the process of labor [34]. This study makes a unique contribution to this line of research using machine learning analysis and a population-based cohort of 124,606 participants. Finally, it is notable that the importance rankings of dental diseases were out of the top 30 in this study. More machine learning investigation is to be done for more concrete evidence and more rigorous validation in this direction.

### 4.3. Limitations

This study had some limitations. Firstly, indicated PTB (preterm birth due to maternal and fetal indication) and spontaneous PTB (preterm birth due to spontaneous preterm labor) have different etiology, but this study could not separate them based on the ICD-10 code. Secondly, this study did not examine possible mediating effects among variables (e.g., the mediating effects of socioeconomic status between heart disease and preterm birth). Thirdly, a recent review suggests that different machine learning approaches would be optimal for different types of data regarding the prediction of preterm birth: the artificial neural network, logistic regression, and/or the random forest for numeric data; the support vector machine for electrohysterogram data; the recurrent neural network for text data; and the convolutional neural network for image data [23]. Uniting various kinds of machine learning approaches for various kinds of PTB data would bring new innovations and deeper insights into this line of research. Fourthly, this study used the default parameters of the random forest, but there exists a possibility of overfitting, and parameter tuning (e.g., the number of trees, their max depth) would help to resolve the issue. Lastly, various dental and gastrointestinal diseases would have different effects on a mother or fetus. However, we did not consider different mechanisms among various diseases in this study.

## 5. Conclusions

In conclusion, this study demonstrated that GERD and infertility are significantly associated with PTB through a valid prediction model for PTB by using machine learning with large-scale population data. Through this study, the need for close surveillance of the obstetric risks as well as the gastrointestinal risk for PTB, which has been overlooked, is ascertained. Further prospective studies to elucidate the pathophysiology of GERD increasing the risk of PTB are needed.

## Figures and Tables

**Figure 1 ijerph-20-01732-f001:**
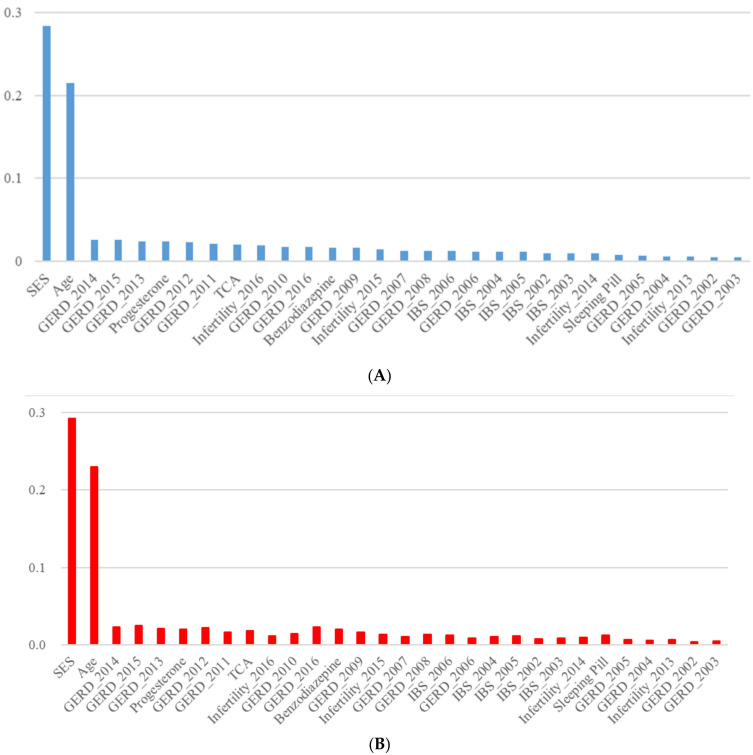
Random forest variable importance-oversampling vs. no sampling for PTB4. (**A**) Oversampling (**B**) No Sampling. Based on random forest variable importance with oversampling data in (**A**), PTB 4 has strong associations with socioeconomic status (0.284), age (0.214), year 2014 GERD (0.026), year 2015 GERD (0.026), year 2013 GERD (0.024), progesterone (0.024), year 2012 GERD (0.023), year 2011 GERD (0.021), tricyclic antidepressant (0.020) and year 2016 infertility (0.019). For example, the accuracy of the model will decrease by 28.4%, 2.6%, or 1.9% if the values of socioeconomic status, year 2014 GERD, or year 2016 infertility are randomly permutated (or shuffled). It should be noted that the random forest variable importance measures for oversampling data were very similar to those for the original data in (**B**). GERD = gastroesophageal reflux disease, IBS = irritable bowel syndrome, SES = Socioeconomic Status, TCA = Tricyclic Antidepressant.

**Table 1 ijerph-20-01732-t001:** Model performance.

Accuracy	PTB 1	PTB 2	PTB 3	PTB 4
No Sampling				
Logistic Regression	0.9827	0.9932	0.9781	0.9772
Random Forest	0.9818	0.9927	0.9767	0.9758
Oversampling				
Logistic Regression	0.5445	0.6025	0.5551	0.5582
Random Forest	0.8599	0.8403	0.8403	0.8403
**AUC**	**PTB 1**	**PTB 2**	**PTB 3**	**PTB 4**
No Sampling				
Logistic Regression	0.5000	0.5000	0.5000	0.5000
Random Forest	0.5022	0.5027	0.5023	0.5023
Oversampling				
Logistic Regression	0.8403	0.6019	0.5539	0.5572
Random Forest	0.8404	0.8404	0.8404	0.8404

**Table 2 ijerph-20-01732-t002:** Random forest variable importance-oversampling.

	PTB1		PTB2		PTB3		PTB4	
1	SES	0.284	SES	0.274	SES	0.281	SES	0.284
2	Age	0.227	Age	0.198	Age	0.218	Age	0.215
3	GERD_2013	0.025	Infertility_2016	0.038	GERD_2014	0.025	GERD_2014	0.026
4	GERD_2015	0.025	Benzodiazepine	0.034	GERD_2015	0.025	GERD_2015	0.026
5	GERD_2012	0.024	GERD_2014	0.025	Progesterone	0.024	GERD_2013	0.024
6	Progesterone	0.023	GERD_2015	0.025	GERD_2012	0.023	Progesterone	0.024
7	TCA	0.022	GERD_2016	0.025	GERD_2013	0.023	GERD_2012	0.023
8	GERD_2014	0.021	GERD_2013	0.024	GERD_2010	0.020	GERD_2011	0.021
9	Benzodiazepine	0.020	GERD_2012	0.022	Benzodiazepine	0.019	TCA	0.020
10	GERD_2011	0.017	GERD_2011	0.021	GERD_2016	0.019	Infertility_2016	0.019
11	GERD_2016	0.016	INFE_2015	0.021	GERD_2011	0.018	GERD_2010	0.017
12	Infertility_2015	0.015	GERD_2010	0.020	Infertility_2016	0.018	GERD_2016	0.017
13	GERD_2008	0.014	Progesterone	0.019	TCA	0.018	Benzodiazepine	0.016
14	GERD_2009	0.014	TCA	0.019	GERD_2009	0.016	GERD_2009	0.016
15	GERD_2010	0.014	GERD_2008	0.013	Infertility_2015	0.013	Infertility_2015	0.014
16	IBS_2006	0.013	GERD_2007	0.012	GERD_2007	0.012	GERD_2007	0.012
17	GERD_2007	0.012	GERD_2009	0.012	IBS_2005	0.012	GERD_2008	0.012
18	IBS_2005	0.012	IBS_2005	0.012	GERD_2008	0.011	IBS_2006	0.012
19	Infertility_2016	0.012	IBS_2006	0.012	IBS_2002	0.010	GERD_2006	0.011
20	GERD_2006	0.010	Sleeping Pill	0.012	IBS_2003	0.010	IBS_2004	0.011
21	IBS_2004	0.010	IBS_2004	0.011	IBS_2004	0.010	IBS_2005	0.011
22	Infertility_2014	0.010	Infertility_2014	0.011	IBS_2006	0.010	IBS_2002	0.009
23	Sleeping Pill	0.010	IBS_2002	0.009	Infertility_2014	0.010	IBS_2003	0.009
24	IBS_2002	0.009	IBS_2003	0.009	Sleeping Pill	0.010	Infertility_2014	0.009
25	IBS_2003	0.009	Infertility_2013	0.008	GERD_2006	0.009	Sleeping Pill	0.008
26	GERD_2005	0.007	GERD_2006	0.007	GERD_2005	0.007	GERD_2005	0.007
27	GERD_2004	0.006	GERD_2004	0.006	GERD_2004	0.006	GERD_2004	0.006
28	Infertility_2013	0.006	GERD_2005	0.006	Infertility_2013	0.006	Infertility_2013	0.006
29	GERD_2002	0.005	Infertility_2012	0.005	GERD_2002	0.005	GERD_2002	0.005
30	GERD_2003	0.005	GERD_2002	0.004	GERD_2003	0.005	GERD_2003	0.005

GERD = gastroesophageal reflux disease, IBS = irritable bowel syndrome, SES = Socioeconomic Status, TCA = Tricyclic Antidepressant.

## Data Availability

The data presented in this study are not publicly available. However, the data are available from the corresponding author upon reasonable request and under the permission of Korea National Health Insurance Service.

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
