# Peer review of "Associations of Preterm Birth with Dental and Gastrointestinal Diseases: Machine Learning Analysis Using National Health Insurance Data"

_ijerph, 2023, doi:10.3390/ijerph20031732_

Round 1

Reviewer 1 Report

The authors of the manuscript " Associations of Preterm ..... Insurance Data" is a very well written manuscript which uses Machine learning to detect preterm birth associated with dental and gastrointestinal diseases.  GERD and periodontal disease association with preterm  using machine learning would help in decreasing the preterm mortality.

Only 2 questions to be answered:

1. In introduction the authors  in line 42-43 used United States economic cost but did not give any cost of Korea since the studies are based in Korea.  So the authors if can change or add an line of econimic burden in Korea should be added.

2. On similar lines in introduction lines (52-53) the periodontal and GERD in United States but no talk about Korea.  If authors can add some information about Korea would be of great help.  Also the different socio-economic difference between US and Korea can be added would clear the affect.

Author Response

Thank you very much for the review of our manuscript. The comments of the reviewers were very constructive and have been used to revise and improve the manuscript.

I wrote the response to reviewer's comment and attached the file.

Reviewer 2 Report

The manuscript presents findings of a large scale study for finding association between preterm birth with dental and gastrointestinal diseases. The positive aspect of the study is the large amount of the data available to the authors which is a key requirement for any machine learning study. The paper is well written (except for a couple of instances) and the methodology is sound. The results described in a presentable manner as well. Overall, the paper requires minor revision for publication subject to addressal of the following comments.

Section 2: It is suggested to include a subsection "Evaluation Metrics" and include definition and formulae for accuracy and AUC (see examples given in 10.32604/iasc.2020.013861)

Section 2.2 : Please provide more details about different parameters of Logistic Regression and Random Forest such as max_depth, n_estimator, max_features etc. 

Please indicate the library used such as Scikit-Learn and its version used in the experiments. Mentioning only Python is not sufficient.

What measures were in place to address overfitting?

Please add details about which library was used for extracting important features?

Section 3 : The authors have identified a number of important features. It will be interesting to comment which of these features are medically relevant and which are medically irrelevant?

Section 5: Please rewrite this section and add directions for future research as well.

English language needs minor edits, below are sample examples

Line 49 : To fully identify .....are involved. [Please rephrase, the word PTB is repeated twice]

Line 126: ... so that the training of machine learning can be ..... [please add "model(s)" after machine learning]

Author Response

We thank you so much for the review of our manuscript. The comments of the reviewers were very constructive and have been used to revise and improve the manuscript.

I wrote the response to reviewer's comment and attached the file.
